# Impact of misclassified defective proviruses on HIV reservoir measurements

**Daniel B. Reeves** [1] ✉, **Christian Gaebler**[2,3], **Thiago Y. Oliveira** [2], **Michael J. Peluso** [4], **Joshua T. Schiffer** [1,5], **Lillian B. Cohn**[1], **Steven G. Deeks** [4] & **Michel C. Nussenzweig** [2,6]

Most proviruses persisting in people living with HIV (PWH) on antiretroviral therapy (ART) are defective. However, rarer intact proviruses almost always reinitiate viral rebound if ART stops. Therefore, assessing therapies to prevent viral rebound hinges on specifically quantifying intact proviruses. We evaluated the same samples from 10 male PWH on ART using the two-probe intact proviral DNA assay (IPDA) and near full length (nfl) Q4PCR. Both assays admitted similar ratios of intact to total HIV DNA, but IPDA found ~40-fold more intact proviruses. Neither assay suggested defective proviruses decay over 10 years. However, the mean intact half-lives were different: 108 months for IPDA and 65 months for Q4PCR. To reconcile this difference, we modeled additional longitudinal IPDA data and showed that decelerating intact decay could arise from very long-lived intact proviruses and/or misclassified defective proviruses: slowly decaying defective proviruses that are intact in IPDA probe locations (estimated up to 5%, in agreement with sequence library based predictions). The model also demonstrates how misclassification can lead to underestimated efficacy of therapies that exclusively reduce intact proviruses. We conclude that sensitive multi-probe assays combined with specific nfl-verified assays would be optimal to document absolute and changing levels of intact HIV proviruses.

Although HIV infection can be suppressed by antiretroviral therapy (ART), latent HIV-1 proviruses persist in the genomes of long-lived CD4+ T cells in people living with HIV[1,2]. Most latent proviruses are defective and cannot replicate[3-7]. However, a small population of intact proviruses are responsible for rebound viremia upon ART interruption[8-10]. Therefore, precise measurements of intact HIV levels are essential to evaluating clinical interventions aimed at altering the dynamics of viral rebound, establishing ART-free HIV remission, and ultimately curing HIV[11-15].

Reservoir size and dynamics have been estimated by a variety of different methods, all with their own advantages and disadvantages in terms of speed, throughput, sensitivity, and specificity to rebound-competent sequences. Quantitative viral outgrowth assays (QVOAs) are the gold standard for identification of rebound-competent sequences, because they measure the number of cells that can be reactivated in vitro to produce infectious virus[4,16]. Multi-probe HIV DNA ddPCR approaches are rapid, scalable, and not biased by in vitro cellular reactivation[17-20]. These assays designate proviral DNA as intact

[1]Vaccine and Infectious Disease Division, Fred Hutchinson Cancer Center, Seattle, WA, USA. [2]Laboratory of Molecular Immunology, The Rockefeller University, New York, NY, USA. [3]Laboratory of Translational Immunology of Viral Infections, Department of Infectious Diseases, Charité -Universitätsmedizin, Berlin, Germany. [4]Division of HIV, Infectious Diseases, and Global Medicine, Department of Medicine, UCSF, San Francisco, CA, USA. [5]Department of Medicine, University of Washington, Seattle, WA, USA. [6]Howard Hughes Medical Institute, The Rockefeller University, New York, NY, USA. ✉e-mail: dreeves@fredhutch.org

or defective based on hybridization of PCR-amplified proviral DNA to probes in two to five relatively conserved locations on the HIV-1 genome. For example, the intact proviral DNA assay (IPDA)[17] targets the HIV-1 Packaging Signal (Ψ) and the Envelope/Rev Responsive Element (RRE). If both regions are amplified within a single provirus, the sequence is said to be intact, and if only one of the two probes is positive the provirus is determined to be defective[17]. Q4PCR performs both quantitative PCR with four different DNA probes and near full-length (nfl) sequence analysis[19]. This approach is far more labor intensive than ddPCR and the relatively inefficient long range amplification admits lower absolute measure of reservoir size compared to ddPCR[21]. However, nfl sequence verification ensures specificity to identify intact sequences[19,22]. Other techniques include single cell omics[23–26], flow cytometry approaches[27,28], enzyme-linked ImmunoSpot[29], and reporter cell-based methods[30].

Most HIV reservoir studies have considered one decade of ART. QVOAs performed on longitudinal samples from that time frame produced a consensus half-life of 4–5 years for the replication-competent latent reservoir[31,32]. On samples from similar time scales, IPDA and Q4PCR also found that intact proviruses decay with a 4–5-year half-lives whereas defective proviral half-lives were much longer with estimates ranging from 100 months to no decay[17,18,33–35] (akin to past estimates of total HIV DNA decay[36,37]).

In a recent study of longitudinal samples collected up to 20 years after ART initiation, IPDA showed a deceleration in intact HIV decay after 7 years[38]. The authors hypothesized that the decelerating decay could arise from shifts in the reservoir toward longer lived HIV-infected CD4 + T cells defined by less-transcriptional activation and/or a more naïve in phenotype. Other recent work shows intact proviruses measured by IPDA may continuously decline but also can decelerate and even increase in some individuals after 10 years of therapy[39].

In addition to understanding long-term dynamics, evaluating therapeutic reductions of HIV reservoirs involves measuring small numbers of intact HIV proviruses, and/or very small changes in these numbers. For these contexts, we quantitatively clarify how assays with

excellent but imperfect specificity for intact proviruses might behave. We developed a mathematical model inclusive of the possibility that some observed intact sequences contain defects outside probe regions such that they are truly defective. Using this model, we highlight how misclassification of defective proviruses could result in underestimates of reservoir reduction in interventional studies.

## Results

### Intact and defective HIV proviral DNA using two assays

To directly compare Q4PCR[19] and IPDA[17] we applied both assays to longitudinal samples obtained from 10 people living with HIV-1 (PWH) on suppressive ART over a period of up to 10 years (Fig. 1A, B, data provided in Supplementary Data 1)[19]. The samples were obtained at time points ranging from 4.5 to 123 months after initiation of ART with a median of 3 (range = [2–4]) samples per participant. IPDA was performed by Accelevir on the same DNA samples used for Q4PCR. Samples with DNA quality or probe amplification issues from IPDA were excluded from further analysis.

Across all samples, IPDA and Q4PCR reported a median of 228 and 6 intact and 2553 and 87 defective proviral copies per million CD4+ T cells, respectively (Fig. 1B). The difference is consistent with prior reports that Q4PCR under-reports HIV DNA due to sequence loss during nfl amplification[22]. In particular, paired IPDA estimates of intact levels were a median of 42-fold greater than Q4PCR, with a wide range [0.2–39000] including a few time points for which Q4PCR estimates were higher.

Some associations were observed among and between assays for HIV DNA levels (correlation coefficients for all comparisons are tabulated in Supplementary Fig. 1). Across all participants and time points, levels of intact and defective proviruses were weakly correlated between Q4PCR and IPDA (Spearman $p = 0.08$ and 0.07 for intact and defective, respectively). Estimates of numbers of intact and defective proviruses were not correlated within each assay, potentially because intact levels change over time whereas defective do not (Spearman $p > 0.13$ for both). However, at early time points (<12 months on ART)

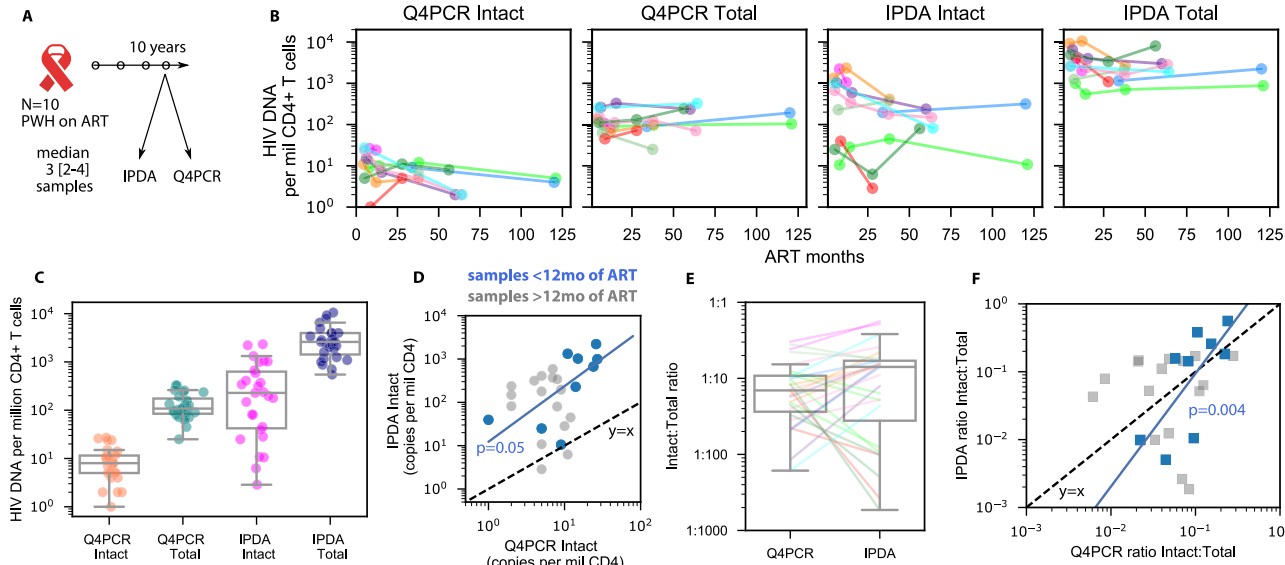

**Fig. 1 | Comparison of reservoir size and half-lives using two intact HIV reservoir assays. A** Study schematic. **B** Longitudinal HIV DNA levels using 2 assays from 10 PWH for intact and total HIV DNA. **C** HIV DNA levels for each assay and proviral category grouped over all $n = 35$ longitudinal time points. Box plots indicate median (center line), interquartile range (box), and 1.5x interquartile range (whiskers). **D** Head-to-head comparison of intact levels measured by both assays. Blue and gray dots indicate samples taken before and after 1 year of ART, respectively. Dashed black line indicates the line $y = x$. Blue line indicates the slope of early time

points, which was significantly correlated (1 sided Spearman $p = 0.05$). **E** Ratio of intact to total HIV DNA for both assays over all $n = 35$ longitudinal time points from $N = 10$ PWH. Box plots indicate median (center line), interquartile range (box), and 1.5x interquartile range (whiskers). **F** Head-to-head comparison of intact:total ratios measured by both assays. Blue and gray dots indicate samples taken before and after 1 year of ART, respectively. Dashed black line indicates the line $y = x$. Blue line indicates the slope of early time points, which was significantly correlated (1 sided Spearman $p = 0.004$).

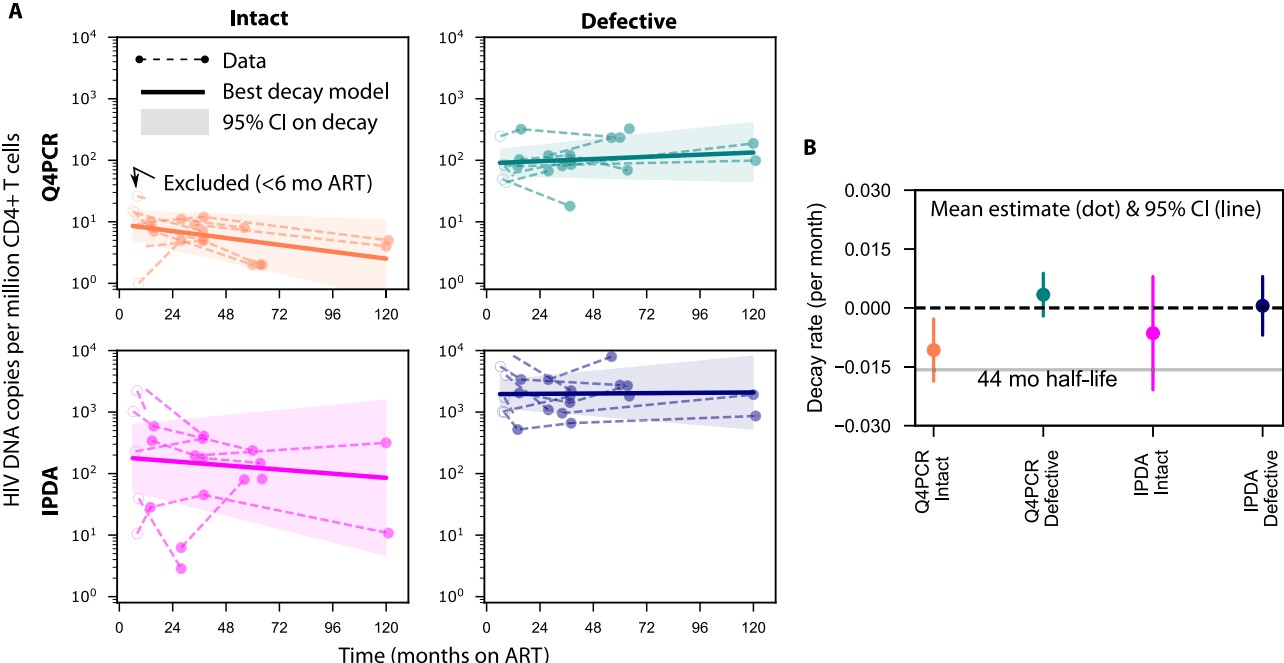

**Fig. 2 | Estimated reservoir decay from two HIV DNA assays. A** Log-linear mixed effects model estimates the average decay (solid line) with confidence band (shaded area) fitted to longitudinal samples from PWH on ART (dots and dashed lines, open dots indicate excluded time points before 6 months). **B** Average (dot) and confidence interval (vertical line) for the estimated decay rate of each assay and proviral category from $N = 10$ PWH. Decay rates are used to allow for zero or positive values (growing population and infinite half-life). A benchmark 44 month half-life is included to contextualize decay rates.

intact reservoir measurements correlated much more strongly between assays ($\rho = 0.8$, $p = 0.004$; Fig. 1D).

### Ratio of intact to total proviruses agrees between assays at early time points

Because the absolute sizes of both intact and defective HIV levels for IPDA were higher, we tested whether the ratio of intact to total HIV DNA could be a conserved quantity for both assays. We found the median ratio was slightly higher for IPDA, but the distributions across all time points were not significantly different, suggesting the ratio of intact to total may agree across assays (Fig. 1E). However, the ratio of intact to total was significantly correlated for early time points but not for all time points (Fig. 1F).

### Estimated reservoir decay using two assays

To assess whether trends for reservoir clearance would agree despite absolute differences in proviral levels, we estimated the decay rate for intact and defective proviruses quantified by each assay. To account for within-individual correlations and different numbers of samples per individual, we used a log-linear mixed effects model. We also accounted for known biphasic decay in proviral levels[34] by only modeling measurements taken after 6 months of ART (Fig. 2A). The model provides a population average for initial size and decay rate with 95% confidence intervals on these estimates (Fig. 2B).

The mean estimated reservoir decay rates indicate that defective proviruses (by either assay) tend not decay (Fig. 2B). Intact decay rates were on average below zero, with average half-lives of 65 and 108 months for Q4PCR and IPDA, respectively. (Note, the estimate of Q4PCR intact half-life here differs slightly from a published estimate using a different analysis method and some of the same data[19]). However, IPDA estimates had more uncertainty such that confidence intervals crossed zero, meaning inclusive of no decay (or increases). Our data at 10 years was sparse, but when we reperformed the estimation excluding data after 6 years of ART, estimates were not qualitatively different.

Detailed examination of prior estimates indicates some intact IPDA half-lives are reported after excluding non-decaying participants (Supplementary Table 1). Therefore, it is not completely clear whether our results here agree with past observations.

### Defective sequences can be misclassified by multi-probe assays and increase observations of intact proviruses

Multi-probe assays that quantify intact proviruses inherently have imperfect specificity. We hypothesized that this imperfect specificity could influence observed levels of intact proviruses in natural and therapeutic settings. To study this hypothesis, we envisioned a system with 3 categories of proviruses: intact, defective, and proviruses which are intact in all IPDA probe locations yet have a defect elsewhere. We term this key third group "misclassified defective" proviruses (Fig. 3A). We encoded this hypothesis into a mathematical model by assuming true intact $I^{true}$ and defective $D$ sequences each have a half-life (or decay rates $\theta_I$ and $\theta_D$). Then, a small fraction of defective sequences $f$ can contribute to observed intact sequences as $I^{obs} = I^{true} + fD$ (Fig. 3B).

Given this model, even if we assume true intact sequences continuously, once $I^{true} < fD$ then the decay rate of observed intact sequences will flatten and converge to the decay rate of misclassified defective proviruses, which decay slowly or not at all. This idea is consistent with the observations that IPDA and Q4PCR measurements diverge with time on ART (Fig. 1). In addition, any therapy that selectively reduces intact proviruses might be underestimated because of the inclusion of misclassified defective proviruses into the observed intact pool. We quantitively explore these phenomena in the next sections.

Misclassified defective sequences might have small and potentially less important defects compared to the general pool of defective proviruses. Therefore, to determine whether the half-life of misclassified defectives differs from the overall pool of defective proviruses, we estimated the half-life of proviruses that were defective by Q4PCR but also amplified by Packaging Signal ($\Psi$) and the envelope Rev Responsive Element (RRE) probes in Q4PCR. The half-life of these

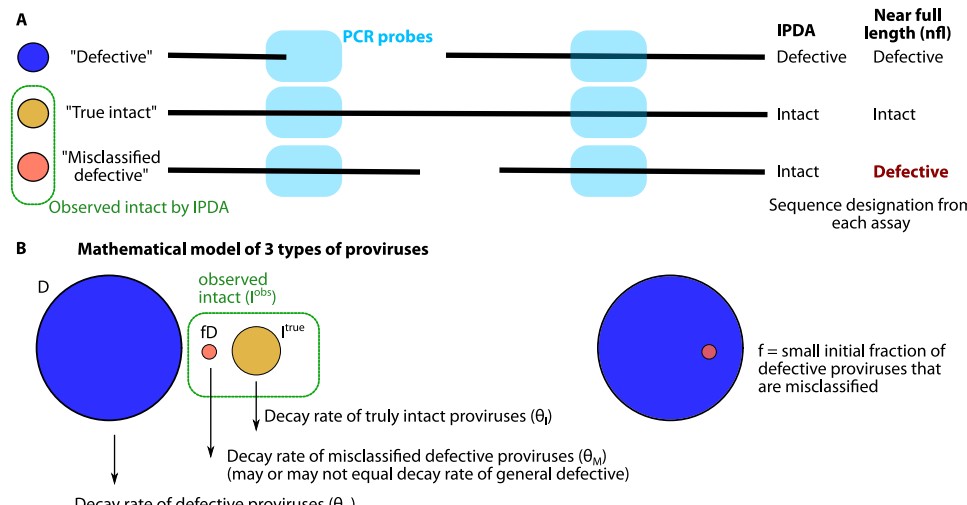

**Fig. 3 | Schematic illustration of misclassified intact proviruses from 2 probe assay. A** Using primer probes targeting 2 regions of the HIV genome, it is possible to have agreement between IPDA and near full length sequencing for intact and defective proviruses. But it is also possible to find misclassified defective sequences: intact in both primer regions but defective elsewhere. **B** Codifying this system into a mathematical model by assuming there are 3 population sizes for the proviral types, each with its own decay rate (which can be zero or growing). Observed intact proviruses are the sum of true intact and misclassified defective proviruses, and misclassified defective proviruses are calculated from the fraction f of all defective proviruses.

PS+ env/RRE+ defective proviruses, which would have been observed as intact by IPDA, was 180 months and not significantly different from overall defective decay rates estimated from either IPDA or Q4PCR (Supplementary Figure 2). This finding helps to simplify the model by letting $\theta_D = \theta_M$.

## The model quantitatively reproduces independent IPDA observations from a 20-year cohort and estimates the misclassification fraction

Misclassification might explain some of the differences between decay rates observed for Q4PCR and IPDA in Fig. 2. However, our data is relatively sparse. Therefore, we assessed whether this model could explain a published larger and longer-duration data set in which intact reservoir deceleration was observed[38]. A successful fit (total R-squared = 0.9) was achieved simultaneously for the experimental levels of observed intact (Fig. 4A) and defective (Fig. 4B) proviruses over time, suggesting this model could reasonably approximate these data.

By fitting this model, we derived average estimates for each of the model parameters in Fig. 3B. We estimated a defective half-life of 320 months and a true intact half-life of 46 months. Both values agree with those using QVOA or nfl sequencing. The estimated misclassification fraction was $f = 0.05$, meaning that, across this population, roughly 5% of defective proviruses may contain defects outside the IPDA probe locations—a result in relatively good agreement with prior in silico sequence analyses[17,18]. With this misclassification fraction and the average initial sizes in the study, we estimate 27% of observed intact proviruses would be misclassified defectives at the initial time points which is nearly identical to the 30% originally reported for the IPDA[17].

## Biological deceleration of intact proviral decay is not mutually exclusive with misclassification

Decelerating decay of intact proviruses could reflect the dynamics of one (or more) longer lived or dividing populations of CD4+ T cells. Our model does not rule out this hypothesis. To examine how additional phases of proviral decay influence our results, we modified our model to include a second population of long-lived intact proviruses ($I_2$, with a half-life of 18.7 years based on the observed long term decay rate of intact proviruses[38], Fig. 4C). We assumed this population is a subset of all intact proviruses, varied its initial value at 1/30, 1/7, and 1/3 of the original intact proviruses, and re-estimated the misclassification

fraction needed to generate observed intact proviral levels. For instance, assuming 1/3 of intact proviruses have an 18.7 year half-life, the model can re-generate observed intact levels correctly provided the misclassification fraction is accordingly reduced from 5% to 1% (Fig. 4D).

## Misclassification results in varying degrees of underestimation for theoretical therapies that specifically reduce intact proviruses

Some proposed HIV cure therapies, such as shock and kill[40,41] or targeted gene therapy[42], might only affect intact proviruses. If intact proviruses are reduced, but misclassified defective proviruses are not, then observed reductions in intact proviruses could be underestimated. To determine theoretical consequences of misclassification, we simulated therapies that reduced true intact proviruses by 2–100 fold. We varied the initial fraction of intact proviruses from 1/100 to 1/2 and the misclassification fraction from 0.5%–8%. Across these simulations, observed intact proviruses did not drop equivalently to true intact proviruses (Fig. 5A, B). Indeed, the simulated true therapeutic efficacy (fold reduction in true intact) was always underestimated by the simulated observed efficacy (fold reduction in observed intact). The misclassification fraction can influence the relationship between true and observed efficacy (Fig. 5C). For low true efficacy (<10 fold reduction), any misclassification fraction could result in near agreement between true and observed efficacy. For higher true efficacy (100-fold reduction) a 0.5% misclassification could still admit agreement between true and observed, depending on initial intact fraction. However, an 8% misclassification admitted at best a 10-fold observed efficacy given a 100-fold true efficacy. In some instances, efficacy could be severely underestimated. The initial fraction of true intact proviruses, a quantity that ranged across individuals with a median of approximately 1/10 (Fig. 1E), plays a large role in how therapeutic efficacy will be observed (Fig. 5D). Regardless of misclassification fraction, if small numbers of intact proviruses disappear, they are not noticed in comparison to the more common misclassified defective proviruses. For example, if intact proviruses are 1/2 of all proviruses, therapeutic efficacy is nearly correctly observed for low efficacy therapies and any misclassification fraction. Yet if 1/1000 of proviruses are intact, even a 0.5% misclassification rate results in an observed therapeutic efficacy of 2, regardless of its true value.

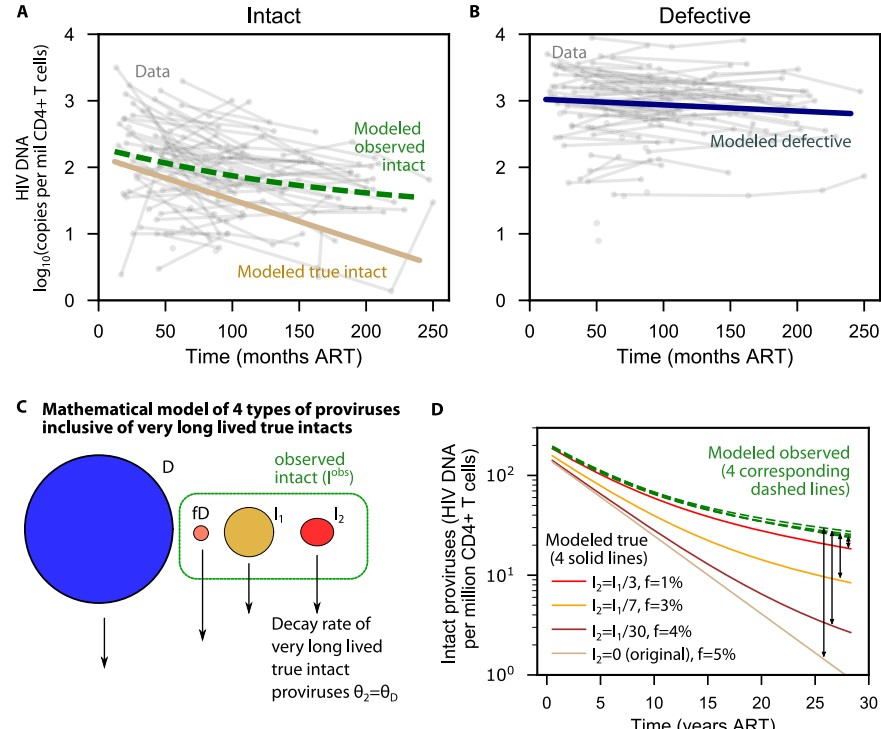

**Fig. 4 | Modeled estimate of intact HIV DNA clearance including misclassification and/or very long lived intact proviruses. A** Observed intact model (dashed green) matches the data (gray) by adding misclassified defectives to true intact proviruses that continuously decay with a 46 month half-life (tan). **B** Defective proviral levels from the model (navy) simultaneously match data (gray) and contribute to long-term decay of observed intact proviruses in (**A**). **C** Mathematical model from Fig. 2 with an additional compartment of truly intact proviruses that are very long lived ($I_2$) and decay with the same rate as defective proviruses. **D** Simulations of this extended model. Each solid line represents an assumption about the initial level of very long lived proviruses (from none, the original assumption, to 1/3 of all truly intact proviruses). A different corresponding misclassification fraction was needed so that all 4 dashed green lines overlay and correctly regenerate observed intact levels from (**A**).

## Balancing sensitivity and specificity in a theoretical analysis of intact provirus assays with additional probe regions

Increasing the number of primer-probe target regions could improve specificity to true intact proviruses and might therefore improve agreement between observed and true intact sequences. Along with nfl sequencing, Q4PCR simultaneously counts proviruses that are intact at 2, 3, and 4 probe regions[22,43], giving us a window into how measurements change when more probes are used to determine intactness. Unfortunately, as the number of probes increased the sensitivity decreased from 4037 to 1044 to 98 proviruses that were 2, 3 or 4 probe positive, respectively. We estimated the mean and 95% confidence interval for the ratio of intact to total proviruses from all participants and all time points for each probe set from our data (Fig. 6A). The percentage of intact proviruses increases with increasing numbers of probes, but the total number of proviruses decreases. Therefore, the statistical uncertainty around the intact fraction increases.

Then, using our mathematical model, we simulated observed intact sequences using theoretical assays including 2, 3, or 4 probes assuming each probe has similar conservation levels as the original IPDA primer-probe sets (Methods)[43]. The theoretical 3 probe assay would decrease misclassification from approximately 5% to 1% and the average (dashed red line Fig. 6B) was projected to have reasonable agreement between observed and true intact provirus levels for approximately 10 years of therapy including near the limit of detection of 10 copies per million CD4+ T cells. The 1.5% rate for the theoretical 3 region assay agreed with an in silico study of more than 1000 sequences and a different published 3 region assay[18], supporting the plausibility of the initial model estimate of 5% for IPDA as well as the theoretical probe

calculation. The theoretical 4 probe assay could lower misclassification further to 0.1%, which was projected to have reasonable agreement between observed and true levels throughout a 20-year study (dashed purple line Fig. 6B). However, when we add uncertainty based on the Q4PCR data in Fig. 6A, the loss of sensitivity for the 3 and 4 probe assays is substantial. The wide confidence intervals mean that changes in intact proviral DNA could also be hard to reliably detect. Therefore, the increase in sensitivity might not compensate for the gains in specificity.

## Discussion

A cure for HIV would benefit millions living with HIV[14,15]. In addition to incremental progress toward cure, any intervention that decreases HIV reservoir size might improve an individual's HIV management by reducing HIV rebound probability after ART pauses[44–48] and could also potentially impact chronic immune activation[49]. Many interventions have been tested, but most do not significantly reduce HIV reservoirs[50]. Going forward, individual interventions that produce small reductions in the intact reservoir might be combined to obtain clinically significant results[14,18]. Therefore, specific assays are crucial to sensitive detection of reservoir reduction and enabling progress towards HIV cure.

Though simple and sensitive, total HIV DNA does not directly measure the rebound competent reservoir because most proviruses are defective[2,4]. On the other hand, quantitative viral outgrowth assays (QVOA) are difficult and potentially biased by in vitro stimulation requirements[31], but specifically count proviruses that could reinitiate infection. Emerging single-cell multi-omics techniques exquisitely illuminate reservoir cell and proviral DNA properties but are also highly complex and their utility for dynamic quantification is not yet

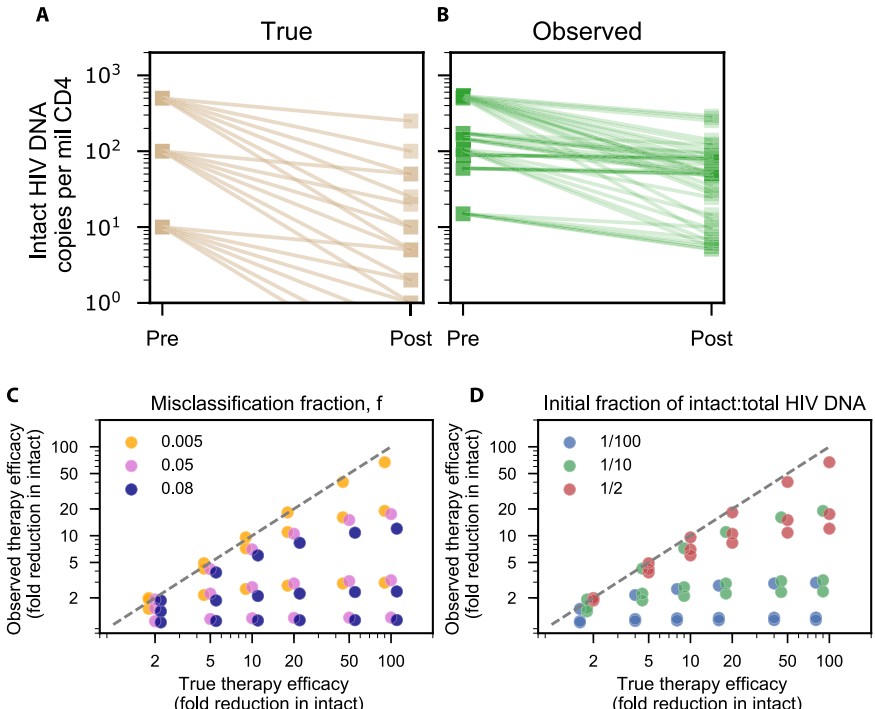

**Fig. 5 | Underestimation of therapeutic efficacy occurs with high true efficacy, high misclassification, and low initial intact fractions.** Simulated true (**A**) vs observed (**B**) reduction in intact HIV DNA after therapies with 6 different efficacies each ranging from 2-fold to 100-fold reduction in intact proviruses (no reduction of defective proviruses) applied to reservoirs with 3 different initial intact fractions, for a total of 18 simulations. **C** True therapeutic efficacy vs observed therapeutic efficacy for the range of therapies, colored by misclassification fraction ranging from 0.5 to 8%. There are 3 dots of each color, for each x-value, representing the variability from the initial fraction. **D** True therapeutic efficacy vs observed therapeutic efficacy for the range of therapies, colored by initial fraction of intact to total HIV DNA ranging from 1/100 to 1/2. There are 3 dots of each color, for each x-value, representing the variability from the misclassification fraction.

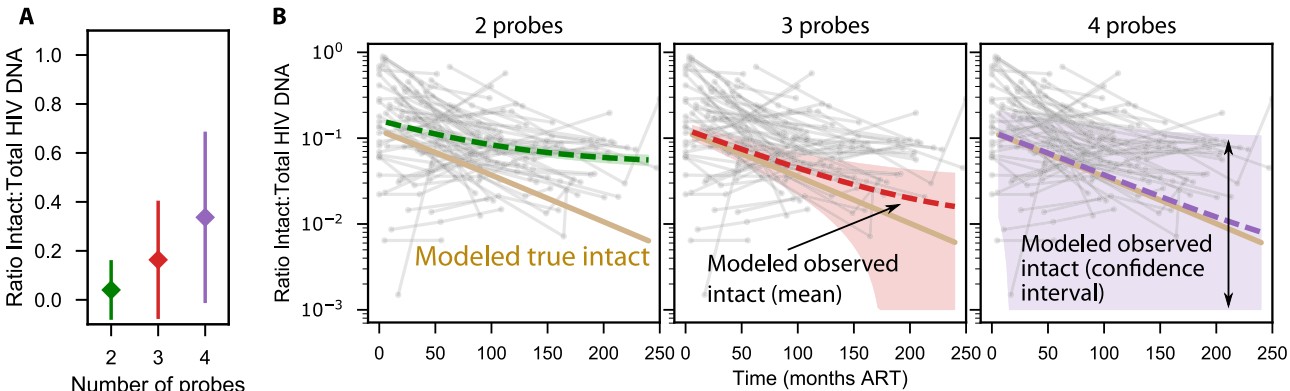

**Fig. 6 | Estimating gains of specificity vs loss of sensitivity in multi-probe assays. A** The mean and confidence intervals for the ratio of intact to total proviruses observed during Q4PCR measurements across all $n = 35$ longitudinal time points from $N = 10$ PWH. **B** Simulating theoretical observed decay given gains in specificity by reducing f (Methods) for 2, 3, and 4 probes. The mean decay (dashed lines) increasingly overlaps with the true decay (solid tan line, identical in each panel) as probes are added. However, additional probes result in fewer detected sequences (see **A**), and confidence interval around the mean decay (shaded area) broadens vastly for more probes.

well studied[23–25]. Based on their commercial availability and simplicity, assays like the intact proviral DNA assay (IPDA) that use two or more PCR probes to define intact HIV sequences have been rapidly adopted as go-to assays for HIV reservoir size in clinical studies[17,18,35,38,51]. Although more challenging than IPDA, assays combining probes with nfl sequencing (e.g., Q4PCR[19,22]) are another emerging technique that reproducibly define intact and defective HIV proviruses.

After applying IPDA and Q4PCR to longitudinal clinical samples from 10 people living with HIV on suppressive ART for 10 years, we found IPDA data admitted a mean intact half-life of 108 months, not significantly distinguishable from defective sequences in these participants, whereas Q4PCR data admitted a mean half-life of 65 months, with confidence intervals not overlapping with zero[31,32]. In addition, IPDA and Q4PCR estimates of intact proviral levels were correlated in observations from the first year of therapy but this correlation disappeared at later time points.

For comparison, in 7 prior studies that used IPDA to measure intact reservoir decay (detailed review in Supplementary Table 1),

mean half-lives ranged from 19-84 months[31,32]. But, throughout these studies, reported means often excluded instances of outlier participants who had long (>100 month) half-lives. Various decay estimation methods were applied, and ART adherence is difficult to confirm but could influence decay. But, evidently, cases of non-decaying intact proviral reservoirs have appeared in the IPDA literature. Further, a relatively large and long-term study (81 participants, some with >12 years on suppressive ART) observed a deceleration in intact decay rate with time on ART[38].

We considered whether some of these findings could be reconciled by considering a third category of HIV proviruses: those which are intact at both IPDA probe locations yet defective elsewhere. These proviruses were acknowledged in original IPDA publications, but their implications for reservoir decay measurements have not been explored[17,52]. Using a mathematical model, we showed that observed deceleration could emerge as true intact sequences decay and what remains in the observed intact pool is slower decaying misclassified defective proviruses. By fitting the model to intact and defective levels, we estimated an average misclassification fraction in this data set was 5%, which generally agrees with estimates derived from libraries of proviral sequences[18].

Notably, our work does not rule out the possibility that intact proviral decay slows over time.

It may be that cells containing intact proviruses persist and undergo clonal expansion and thereby contribute to decelerating decay or even increases of intact proviruses in individuals on long-term therapy[39]. Our simulations including an additional long-lived compartment could also generate observed data, provided the estimated misclassification rate was lowered. Additional prolonged longitudinal studies with more specific assays such as QVOA or nfl sequencing will illuminate novel decay dynamics and the biology of the latent reservoir after many years of ART.

Notwithstanding changes in long-term biological decay, we also showed that misclassification leads to underestimates of efficacy from therapies that specifically target intact proviruses. This phenomenon is particularly important to consider for trial participants having low fractions of intact:total HIV DNA.

Assays adding additional primer-probe target regions could improve specificity, but as usual a balance between specificity and sensitivity emerges. Q4PCR includes sequence information but underestimates absolute levels of intact proviruses relative to IPDA[21,43]. For now, 2 probes appear to achieve a pragmatic balance, and if next generation multi-probe assays are designed, they should be optimized using proviral libraries to limit misclassification (specificity) and maximize amplification (sensitivity).

In epidemiology, it is well known that diagnostic assays with excellent specificities (>90%) will still create a high number of false positives when applied to a low-prevalence population for a given disease. We suggest that this phenomenon is relevant for applying the IPDA to HIV DNA.

Our modeling is inherently theoretical, and limitations include that the misclassification rate itself is assumed to be constant across individuals, which may not be reasonable given the massive diversity of HIV and the clonality of HIV reservoirs[9,10,19,53–57]. Although we focused on IPDA overestimation of intact sequences, in some unusual cases IPDA apparently underestimated replication competent sequences because of probe mismatches to intact sequences[58].

In conclusion, multi-probe ddPCR assays are a pragmatic middle ground with excellent sensitivity and high but imperfect specificity for quantifying intact proviruses. Going forward, care must be taken so that misclassified defective sequences that ultimately emerge from imperfect specificity do not obscure changes in intact reservoir sizes. When it is crucial to detect changes during interventional studies, assays that include additional probes and/or sequence verification are a natural solution.

## Methods

### Human participants

This study was approved by the institutional review board of the NIH National Institute of Allergy and Infectious Diseases and registered on ClinicalTrials.gov: NCT00039689. All study participants provided written informed consent, and leukapheresis products were collected in accordance with the protocol. All study participants were of male sex based on self-reporting. 5 participants were White, 3 Black and 2 Unknown/Hispanic-Latino. Median age at last study time-point 42 years (range 27–61). 8 participants initiated antiretroviral therapy after chronic infection, 2 participants initiated antiretroviral therapy after early infection. Study participants were enrolled at the Clinical Center, National Institute of Allergy and Infectious Diseases, National Institutes of Health, Bethesda, MD 20892, USA. Characteristics of study participants including sex and gender can be found in Table S1 of Ref. [19].

### Q4PCR measurements

Q4PCR was performed as previously described[21,39]. Briefly, genomic DNA from 1 to 5 million total CD4 T cells was isolated using the Gentra Puregene cell kit (Qiagen) or phenol-chloroform, and the DNA concentration was measured using a Qubit high-sensitivity kit (Thermo Fisher Scientific). Next, an outer PCR (nfl1) was performed on genomic DNA at a single-copy dilution using outer PCR primers BLOuterF (5′-AAATCTCTAGCAGTGGCGCCCGAACAG-3′) and BLOuterR (5′-TGAGGGATCTCTAGTTACCAGAGTC-3′)[37]. Undiluted 1 μl aliquots of the nfl1 PCR product were subjected to a Q4PCR reaction using a combination of four primer/probe sets that target conserved regions in the HIV-1 genome. Each primer/probe set consists of a forward and reverse primer pair as well as a fluorescently labeled internal hydrolysis probe as previously described:[21] PS forward, 5′-TCTCTCGACGCAGGACTC-3′; PS reverse, 5′-TCTAGCCTCCGCTAGTCAAA-3′; PS probe, 5′-/Cy5/TTTGGCGTA/ TAO/CTCACCAGTCGCC-3′/IAbRQSp (Integrated DNA Technologies); env forward, 5′-AGTGGTGCAGAGAGAAAAAAGAGC-3′; env reverse, 5′-GTCTGGCCTGTACCGTCAGC-3′; env probe, 5′-/VIC/ CCTTGGGGTTCTTGGGA-3′/MGB (Thermo Fisher Scientific); gag forward, 5′-ATGTTTTCAGCATTATCAGAAGGA-3′; gag reverse, 5′- TG CTTGATGTCCCCCCACT-3′; gag probe, 5′-/6-FAM/CCACCCC- AC/ZEN/ AAGATTTAAACACCATGCTAA-3′/IABkFQ (Integrated DNA Technologies); and pol forward, 5′-GCACTTTAAATTTTCCCATTAGTCCTA-3′; pol reverse, 5′-CAAATTTCTACTAATGCTTTTATTTTTTC-3′; pol probe, 5′-/NED/AAGCCAGGAATGGA-TGGCC-3′/MGB (Thermo Fisher Scientific).

Each Q4PCR reaction was performed in a 10 μl total reaction volume containing 5 μl TaqMan universal PCR master mix containing Rox (catalog no. 4304437; Applied Biosystems), 1 μl diluted genomic DNA, nuclease-free water, and the following primer and probe concentrations: PS, 675 nM forward and reverse primers with 187.5 nM PS internal probe; env, 90 nM forward and reverse primers with 25 nM env internal probe; gag, 337.5 nM forward and reverse primers with 93.75 nM gag internal probe; and pol, 675 nM forward and reverse primers with 187.5 nM pol internal probe.

qPCR conditions were 94 °C for 10 min, 40 cycles of 94 °C for 15 s, and 60 °C for 60 s. All qPCRs were performed in a 384-well plate format using the Applied Biosystem QuantStudio 6 or 7 Flex real-time PCR system. qPCR data analysis was performed as previously described[21] using ThermoFisher Design and Analysis Software 2.4.3.

Generally, samples showing reactivity with two or more of the four qPCR probes were selected for a nested PCR (nfl2). This focused approach detects the subset of defective proviruses that are positive for at least 2 of the 4 oligonucleotide probes which in turn leads to an underestimation of the absolute number of defective proviruses. The nfl2 reaction was performed on undiluted 1 μl aliquots of the nfl1 PCR product. Reactions were performed in a 20 μl reaction volume using Platinum Taq high-fidelity polymerase (Thermo Fisher Scientific) and PCR primers 275 F (5′-ACAGGGACCTGAAAGCGAAAG-3′) and

280 R (5′-CTAGTTACCAGAGTCACACAACAGACG-3′)[37] at a concentration of 800 nM. Library preparation and sequencing were performed as previously described[21]. Sampling time points without recovery of intact sequences were determined at lower limit of detection for statistical analyses. Limit of detection is calculated as half of the proviral frequency assuming 1 intact sequence out of the highest input of sampled cells.

## IPDA measurements
Aliquots of genomic DNA samples that were used for Q4PCR measurements were sent to Accelevir Diagnostics who performed IPDA measurements as previously described[51]. DNA quality was determined by spectrophotometry. DNA samples that did not pass company quality control or showed probe amplification issues were flagged by Accelevir Diagnostics and were excluded from further analysis.

## Estimating decay half-lives from longitudinal measurements of population sizes
To estimate decay half-lives, we apply a log-linear mixed effects model implemented using the statsmodels package in Python. Using the data in Supplementary Data 1, we restricted to time points after 6 months of ART to avoid early rapid (biphasic) decays as observed previously[34,53]. We then estimate the decay rate ($k$) and the initial value (y-intercept) as well as confidence intervals on the decay rate for each assay (Q4PCR and IPDA) and each proviral type (intact and defective). We then calculate half-lives using the decay rate $hl = \ln(2)/-k$, where positive decay rates indicate proviral increases such that the half-life would be formally infinite, and a doubling time would need to be defined instead.

## Simulating reservoirs including misclassified defective proviruses
To simulate HIV reservoirs including misclassified defective proviruses, we developed a simple mathematical model in which the dynamics of true intact $I^{true}$ and defective $D^{true}$ proviruses each follow exponential decays with initial sizes and decay rates ($\theta_I$ and $\theta_D$) drawn for each individual $j$. Thus,

$$I_j^{true}(t) = I_j^{true}(0)\exp(-\theta_{I,j}t), \tag{1}$$

and similarly, $D_j^{true}(t) = D(0)\exp(-\theta_{D,j}t)$. The probability that a putatively intact sequence is truly defective, that is the chance that the sequence is intact for all probes yet has a defect elsewhere, is modeled with the misclassification rate $f$ that is the same for all individuals. Thus, the observed intact sequences in the $j$-th individual follows

$$I_j^{obs}(t) = I_j^{true}(t) + fD_j^{true}(t), \tag{2}$$

and similarly, $D_j^{obs}(t) = D_j^{true}(t) - fD_j^{true}(t)$.

## Fitting the misclassification model to experimental data
We used a least-squares approach and SciPy's optimize package to fit to both intact and defective HIV proviral levels simultaneously against the misclassification model. The score function to be optimized was the root-mean-squared expression

$$RMS = \left[\sum_{t,j}\left(\log_{10}I_j^{obs}(t) - Y_j(t)\right)^2 + \left(\log_{10}D_j^{obs}(t) - Z_j(t)\right)^2\right]^{1/2}, \tag{3}$$

where $Y$ and $Z$ are log10 transformed experimental observations from each individual. Fitting on the log-scale implies that noise is log-normally distributed and means score accuracy is balanced across the two data types despite their large differences on the absolute scale. Five parameters were estimated: the initial levels of intact and defective proviruses, the two half-lives, and the misclassification rate.

Identifiability was confirmed by an additional analysis using population nonlinear mixed effects modeling in Monolix[59] which admitted similar estimates for parameters as well as a lack of correlations among population parameters and residual squared errors below 50%.

## Simulating reservoirs including additional decay phase for intact proviruses
Based on the possibility that intact proviruses do not decay indefinitely with the same rate, we re-simulated observed intact proviral levels using a model with multiple phases of true intact proviral decay. In this case

$$I^{true}(t) = I_1\exp(-\theta_I t) + I_2\exp(-\theta_{I2}t) \tag{4}$$

Where $I_2$ and $\theta_{I2}$ represent the initial size (computed as a fraction of $I_1$) and the decay rate of the second phase, respectively. For correspondence with the initial measurement of the second phase decay, we imputed an 18.7 year half-life for $\theta_{I2}$. Observed intact levels are again calculated using Eq. (2) Because of the extra contribution to the true intact pool at later times, the as $I_2$ grows, the misclassification fraction $f$ must be reduced to maintain a match to observed data.

## Calculating loss of sensitivity from Q4PCR with multiple probes
Q4PCR determines proviral intactness with both near full length sequencing and 2-4 probes. However, the requirement to bind to more probes results in a loss of sequences. For 2, 3, and 4, probes there were 270:3676, 187:857, and 49:49 intact to defective sequences quantified. Many participants had missing time points and time points with zero intact sequences. Therefore, we estimated a coarse average ratio $r_{I:D}$ of intact to total sequences and a 95% binomial confidence interval on this ratio ($CI = 1.96\sqrt{\frac{r_{I:D}(1-r_{I:D})}{n}}$) where $n$ is the total number of sequences. This interval was then applied to model output to demonstrate how increasing probes raises specificity but at a cost to sensitivity (Fig. 6).

## Extrapolating misclassification rates to assays including more ddPCR probes
The probability of a misclassification can be restated as the probability $p_i$ that a defective primer probe at region $i$ ensures a completely defective HIV sequence. With this definition, the probability of finding a defective sequence which is intact at both sites of a two-probe assay but defective elsewhere is

$$f = (1-p_i)(1-p_j) \tag{5}$$

If we assume both probes have similar probabilities, i.e., $p_1 = p_2 = p$, then we can input our estimated average misclassification rate $f \sim 5\%$ and find $p \sim 78\%$–that a defective read at each target region is likely to properly call a defective sequence in roughly 3/4 HIV-1 proviruses. Therefore, if it were possible to develop assays with $x$ probes having the same level of conservation, the total misclassification rate would be $f = (1-p)^x$. When we simulate 3 and 4 probe assays, the misclassification fractions are accordingly $f = 1\%$ and 0.2%, respectively. Using these misclassification fractions, we simulated the observed intact proviruses using Eq. (2), and added confidence intervals calculated from the Q4PCR data.

## Reporting summary
Further information on research design is available in the Nature Portfolio Reporting Summary linked to this article.

## Data availability

All data used to generate all figures is available in Supplementary Data 1. Viral sequences have been deposited in GenBank with the accession codes OQ948507-OQ953743.

## Code availability

All custom plotting scripts and code used for modeling and to generate all figures are freely available at https://github.com/dbrvs/IPDAmodel/releases/tag/v1.

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

## Acknowledgements

The authors are extremely grateful to the study participants who devoted their time and bodies to our research and the Rockefeller University Hospital Research support office and nursing staff for help conducting the study. We also thank A Chakraborty for early reading and comments. This work was supported by the National Institutes of Health (K25 AI155224 to D.B.R., K23 A137522 to M.J.P., UM1 AI100663 and R01AI129795 to M.C.N.); the UW/Fred Hutch Center for AIDS Research (P30 AI027757). the Einstein-Rockefeller-CUNY Center for AIDS Research (1P30 AI124414-01A1); and the Robertson Fund. C.G. was supported by the HJH-Foundation, the Robert S. Wennett post-doctoral fellowship, the Shapiro-Silverberg Fund for the Advancement of Translational Research, and the National Center for Advancing Translational Sciences (UL1 TR001866). L.B.C. is supported by Delaney AIDS Research Enterprise (DARE) Collaboratory (UM1AI126611) and REACH: Research Enterprise to Advance a Cure for HIV (UM1AI164565). M.C.N. is a Howard Hughes Medical Institute (HHMI) Investigator, and this article is subject to HHMI's Open Access to Publications policy.

## Author contributions

M.C.N., L.B.C., C.G., and D.B.R. conceived the project. D.B.R. performed all modeling, analysis, and generated all figures. T.Y.O. and M.J.P. pro-vided data. M.C.N., S.G.D., and J.T.S. supervised the project. D.B.R. and M.C.N. wrote the manuscript and all authors revised.

## Competing interests

The authors declare no competing interests.
