## [Peer Review File · Nature Communications]

Impact of misclassified defective proviruses on HIV reservoir measurementsREVIEWER COMMENTS

Reviewer #1 (Remarks to the Author):

The paper of Reeves from the Nussenzweig group is timely. It addresses points in the ongoing debate whether or not primer based assessment of intactness is appropriate and can be applied for clinical trials. The field of HIV cure has evolved to more in depth evaluation of the viral reservoir. There are some concerns concerning the applicability of the conclusions. Additional analysis is needed to make sure that the data supports the conclusions and claims as explained in the concerns and comments.

Major comments:

1. The methodology section is incomplete, it's important to see how the Q4PCR/IPDA were exactly performed and to see the raw data.

Does the Supplementary Table 1 list total (?) number of cells observed at each timepoint. How were those numbers divided by each assay?

2. Given that Q4PCR yields qualitative information on the suspected qPCR positive wells, one could look at those generated genomes. Where all sequenced wells intact. This could give further insights why IPDA misclassifies or not. The authors could screen for Q4PCR sequences with perfect match to IPDA primer/probe sets but still defective elsewhere so called 'misclassified defective proviruses'. This would give the actual estimate of this fraction across the 10 patients based on Q4PCR data.

3. What type of defects were observed, deletions/frameshifts/stop-codons? Missing information but could add something?

4. How exactly were samples excluded from the cohort? (line 120) Based on this, hard to check what actual drop-out rate. If probe amplification issues due to primers not binding would arise, you would suspect all IPDA readouts for that participant would fail which is not the case? I count 3 individuals with some issues on at least 1 timepoint but some other timepoints are still used?

5. Line 219: How was this assumption made? The real value observed in the Q4PCR to indicate the context on how conservative/reliable this assumption of 0.01 is (In my opinion this is rather conservative).

6. Line 268: it is rather easy to simulate but is it realistic to find additional probe sets with similar conservation levels that are spread so it's practically possible to run multiplex assay on ddPCR? In addition, others have shown issues with detection levels/drop outs of some region (see White paper which shows some Q4PCR probes more prone to dropout).

7. Line 333: The authors give a lot of importance to the paper ref 38. The number of people reaching the 15y timepoint is about max 15 and the number of people reaching the 20y timepoint is 2. So the mean follow up is rather 12y with some exceptional cases reaching 20y. The text is misleading.

Minor comments:

1. Line 23: does IPDA have really a superior sensitivity for intact? It has a superior sensitivity to detect a signal which might be overestimated to be intact.
2. Line 52. Other assay do exist that also measure HIV-1 reservoir size, such as flow-based methods (eg HIV-flow and STIP-Seq)
3. Through the text, inconsistent use of the term near full-length (some examples: line 70 72, 78).

Reviewer #2 (Remarks to the Author):

In the manuscript "Impact of misclassification of intact HIV sequences on longitudinal reservoir measurements" from Reeves et al., the Authors compare the performances of intact proviral DNA assay (IPDA) and near-full length Q4PCR, for the detection and evaluation of the amount of intact and defective HIV genomes in patients living with HIV or under Anti-Retroviral Therapy (ART). From this comparison the Authors found that IPDA was significantly more sensitive than Q4PCR, which however, the Authors claim, is the result of an imperfect specificity of IPDA. To support this claim, the Authors developed a mathematical model which show that indeed IPDA apparently misclassify the defective HIV genomes. The authors show (by modeling) that the inclusion of 3 to 5 probes would allow to better measure the latent reservoir of intact HIV-1 proviruses.

Comments

This is an interesting and well written manuscript that shows the limitations of current intact proviral DNA assay through mathematical modelling. However, although the mathematical modeling is convincing, the conclusions remain purely theoretical. To fully sustain their claims, the Authors should (as they suggest) use multi-probe ddPCR assays (for example 5 probes). Without this experimental validation, the manuscript would remain very specialistic, and thus it would be more suited for a more specialized journal. On the other hand, experimental validation of a novel multi probe ddPCR assay, as proposed by the Authors, would result in an important gold standard in the field and well suited for publication in Nature Communications. I think that the request is fair as the Authors have the samples and, of course, the know how to perform such experimental validation.

We thank the reviewers for their evident thoughtful read of our manuscript. Here we respond to all comments in a point by point manner. There are a few small yellow highlights in the manuscript indicating specific minor changes. Note also that the results section is heavily revised based on these excellent suggestions, with several new analyses included into the new Figs 4, 5, and 6.

Reviewer #1 (Remarks to the Author):

The paper of Reeves from the Nussenzweig group is timely. It addresses points in the ongoing debate whether or not primer based assessment of intactness is appropriate and can be applied for clinical trials. The field of HIV cure has evolved to more in depth evaluation of the viral reservoir. There are some concerns concerning the applicability of the conclusions. Additional analysis is needed to make sure that the data supports the conclusions and claims as explained in the concerns and comments.

Major comments:

1. The methodology section is incomplete, it's important to see how the Q4PCR/IPDA were exactly performed and to see the raw data.

Thanks for pointing this out. We have added Q4PCR and IPDA to the Methods section. In addition, we have uploaded viral sequences to GenBank (accession codes pending) and included a Supp Table 1 that contains all raw data for each participant.

Does the Supplementary Table 1 list total (?) number of cells observed at each timepoint. How where those numbers divided by each assay?

Thanks for this clarification, we now provided a more detailed updated Supp Table 1.

2. Given that Q4PCR yields qualitative information on the suspected qPCR positive wells, one could look at those generated genomes. Where all sequenced wells intact. This could give further insights why IPDA misclassifies or not. The authors could screen for Q4PCR sequences with perfect match to IPDA primer/probe sets but still defective elsewhere so called 'misclassified defective proviruses'. This would give the actual estimate of this fraction across the 10 patients based on Q4PCR data.

This is a very good point. First, the genomic signatures of IPDA vs Q4PCR have been documented in prior manuscripts (Gaebler et al. JCI and White et al. PPath for instance).

Second, with respect to screening for matches, we thought this was an excellent idea. When we screened the proviral sequences from these participants in the current study, of 5,237 total proviral sequences 422 would have a perfect match in both IPDA primer/probe sets, the majority of which actually would have a defect elsewhere. Previously we predicted that 66% of intact sequences from the Los Alamos database would be identified by PS+env based on looser (allowing for non-exact match) binding criteria. Using these loose criteria on the present data showed 17% of two-probe intact viruses did not have any defects elsewhere. However, it's worth noting that in-silico primer/probe signal predictions are imperfect. For that reason, we performed extensive signal pattern and sequence polymorphism analyses based on actual Q4PCR primer/probe signal combinations and associated proviral genomes from individual participants living with HIV in the past (Gaebler et al. JEM and JVI papers). We observed that the precision of the IPDA PS and env probes to identify intact proviruses varied

substantially across individuals (Gaebler et al. JVI). While primer/probe signal and sequencing results from Q4PCR provide some insights into assay performance of the PS+env primer/probe combination, sequencing of proviruses in IPDA PS+env double positive droplets would ultimately be needed to confirm this observation.

Finally, in the new manuscript (Fig 6) we now also directly show from Q4PCR what fraction would be called intact by 2+, 3+ and 4 probes and calculate a confidence interval on this estimate. This analysis shows conclusively how increases in specificity engender decreases in sensitivity (more uncertainty).

3. What type of defects were observed, deletions/frameshifts/stop-codons? Missing information but could add something?

Thanks, good suggestion, additional information is now included in Supp Table 1.

4. How exactly were samples excluded from the cohort? (line 120) Based on this, hard to check what actual drop-out rate. If probe amplification issues due to primers not binding would arise, you would suspect all IPDA readouts for that participant would fail which is not the case? I count 3 individuals with some issues on at least 1 timepoint but some other timepoints are still used?

DNA samples that did not pass company quality control or showed probe amplification issues were flagged by Accelevir Diagnostics and were excluded from further analysis (see Suppl Table 1). According to Accelevir's assessment, a probe amplification issue occurred at the 60 month time point for participant 8 while prior measurements did not fail. We agree that probe amplification issues due to primer/probe mismatches are expected to be consistent across different measurements. However, there is a 32 month interval between the second and third timepoint and reservoir clonal dynamics might contribute to the observed discrepancy in primer/probe amplification.

5. Line 219: How was this assumption made? The real value observed in the Q4PCR to indicate the context on how conservative/reliable this assumption of 0.01 is (In my opinion this is rather conservative).

Thanks for bringing this up. We had originally chosen a number that was simply a round number and one we thought was conservative (1%). However, from these reviews and other internal reviews, this exercise actually seemed to add more confusion than insight. Therefore, in the new version we removed this analysis in favor of the analysis in which we estimate this parameter (finding ~5%) from the other dataset and also this opened up room for the new analyses/figures (Fig 4-6).

6. Line 268: it is rather easy to simulate but is it realistic to find additional probe sets with similar conservation levels that are spread so it's practically possible to run multiplex assay on ddPCR? In addition, other have shown issues with detection levels/drop outs of some region (see White paper which show some Q4PCR probes more prone to dropout).

Thanks, yes, we do think the difficulty of finding these probe sets is part of the justification for the theoretical analysis. Actually, Reviewer 2 was hoping we could validate an assay with more probes. In light of this we attempted to do a new analysis looking at multi-probe Q4PCR data. The analysis shows that increasing specificity with more probes would also predictably decrease sensitivity (Fig 5).

7. Line 333: The authors give a lot of importance to the paper ref 38. The number of people reaching the 15y timepoint is about max 15 and the number of people reaching the 20y timepoint is 2. So the mean follow up is rather 12y with some exceptional cases reaching 20y. The text is misleading.

Excellent points, in the new version we have corrected the wording about this study and have lessened the importance of this analysis as it is one particular example of implications of misclassification. We also now include Fig 6 as another example: that of measuring therapeutic efficacy in light of misclassification – an issue that was discussed but not actually modeled in the original version.

Minor comments:

1. Line 23: does IPDA have really a superior sensitivity for intact? It has a superior sensitivity to detect a signal which might be overestimated to be intact.

Interesting point. We agree and have included a light sentence in the discussion to this effect. However, we also being relatively careful in our critique of the IDPA assay to make sure that our pointing out misclassification issue is not construed as a dismissal of the assay's utility.

2. Line 52. Other assay do exist that also measure HIV-1 reservoir size, such as flow-based methods (eg HIV-flow and STIP-Seq)

Apologies, we now mention and reference these assays in the introduction.

3. Through the text, inconsistent use of the term near full-length (some examples: line 70 72, 78).

Excellent catch thanks. This is revised.

Reviewer #2 (Remarks to the Author):

In the manuscript "Impact of misclassification of intact HIV sequences on longitudinal reservoir measurements" from Reeves et al., the Authors compare the performances of intact proviral DNA assay (IPDA) and near-full length Q4PCR, for the detection and evaluation of the amount of intact and defective HIV genomes in patients living with HIV or under Anti-Retroviral Therapy (ART). From this comparison the Authors found that IPDA was significantly more sensitive than Q4PCR, which however, the Authors claim, is the result of an imperfect specificity of IPDA. To support this claim, the Authors developed a mathematical model which show that indeed IPDA apparently misclassify the defective HIV genomes. The authors show (by modeling) that the inclusion of 3 to 5 probes would allow to better measure the latent reservoir of intact HIV-1 proviruses.

This is an interesting and well written manuscript that shows the limitations of current intact proviral DNA assay through mathematical modelling. However, although the mathematical modeling is convincing, the conclusions remain purely theoretical. To fully sustain their claims, the Authors should (as they suggest) use multi-probe ddPCR assays (for example 5 probes). Without this experimental validation, the manuscript would remain very specialistic, and thus it would be more suited for a more specialized journal. On the other hand, experimental validation

of a novel multi probe ddPCR assay, as proposed by the Authors, would result in an important gold standard in the field and well suited for publication in Nature Communications. I think that the request is fair as the Authors have the samples and, of course, the know how to perform such experimental validation.

We are grateful for the read and the suggestion from this reviewer as it illuminated an important clarifying analysis on sensitivity (vs specificity) of multiprobe assays.

First, we apologize because the section in our manuscript on theoretical aspects of multi-probe use was insufficiently clear so has been rewritten. Second, as noted by reviewer 1, finding conserved primer-probe sets is exceedingly difficult in reality. Moreover, commercially available platforms for multi-color probe analysis are not currently available to make these assays scalable. Therefore, we are unable to honor the reviewer request exactly.

However, we believe we honored the scientific point underlying this request. Q4PCR inherently has the capability to examine intact sequences with additional (from 2 up to 4) probes. Based on the reviewer's suggestion, we now explicitly include data on intact proviruses quantified by these probe sets into the manuscript. The data from these participants was not granular enough to estimate half-lives for each probe set explicitly, but it illustrates a crucial point about how increasing specificity through increasing probe sets will predictably reduce sensitivity.

We could then show explicitly using these data in our model (new Fig 6), that even though the average trajectory of intact proviruses measured by a more probe assay could match more closely to the true trajectory, there would be a lot of uncertainty around the average in reality. Therefore, a much larger population of participants would be required to begin to approximate this average and correctly estimate the long-term decay rate (or change after a therapeutic reduction). This analysis was relatively convincing to us that 2 probes is a reasonable compromise, and that full length sequencing is, given the existing assay now, an easier way to ensure specificity.

REVIEWERS' COMMENTS

Reviewer #1 (Remarks to the Author):

All my comments have been well addressed, therefore I consider the manuscript mature enough for publication in Nat Comm

Reviewer #2 (Remarks to the Author):

The Authors have now included additional experimental data and analyses well supporting their claims.

All requests have been honored.